# Innovative Topical Patches for Non-Melanoma Skin Cancer: Current Challenges and Key Formulation Considerations

**DOI:** 10.3390/pharmaceutics15112577

**Published:** 2023-11-03

**Authors:** Sangseo Kim, Candace M. Day, Yunmei Song, Amy Holmes, Sanjay Garg

**Affiliations:** Centre for Pharmaceutical Innovation, Clinical and Health Sciences, University of South Australia, Adelaide, SA 5000, Australia; sangseo.kim@mymail.unisa.edu.au (S.K.); candace.day@unisa.edu.au (C.M.D.); may.song@unisa.edu.au (Y.S.); amyholmes474@gmail.com (A.H.)

**Keywords:** non-melanoma skin cancer, topical patches, drug delivery, formulation, microneedles, nanotechnology

## Abstract

Non-melanoma skin cancer (NMSC) is the most prevalent malignancy worldwide, with approximately 6.3 million new cases worldwide in 2019. One of the key management strategies for NMSC is a topical treatment usually utilised for localised and early-stage disease owing to its non-invasive nature. However, the efficacy of topical agents is often hindered by poor drug penetration and patient adherence. Therefore, various research groups have employed advanced drug delivery systems, including topical patches to overcome the problem of conventional topical treatments. This review begins with an overview of NMSC as well as the current landscape of topical treatments for NMSC, specifically focusing on the emerging technology of topical patches. A detailed discussion of their potential to overcome the limitations of existing therapies will then follow. Most importantly, to the best of our knowledge, this work unprecedentedly combines and discusses all the current advancements in innovative topical patches for the treatment of NMSC. In addition to this, the authors present our insights into the key considerations and emerging trends in the construction of these advanced topical patches. This review is meant for researchers and clinicians to consider utilising advanced topical patch systems in research and clinical trials toward localised interventions of NMSC.

## 1. Introduction

Non-melanoma skin cancer (NMSC)—primarily composed of basal cell carcinoma (BCC) and squamous cell carcinoma (SCC)—is the most prevalent cancer type worldwide, accounting for approximately one-third of all cancer diagnoses each year [1]. The global incidence of NMSC is increasing, with an estimation of 6.3 million new cases worldwide in 2019, and is anticipated to increase further, resulting in a huge burden on the healthcare system [2]. NMSC is generally found on body areas with high sun exposure, underscoring the significant role of ultraviolet (UV) radiation in their aetiology [3]. Other risk factors for NMSC also include fair skin, immune suppression, previous exposure to radiation, human immunodeficiency virus, and human papillomavirus and smoking, among others [4,5].

The current management strategies for NMSC span a range of options, including surgical interventions, radiation therapy, systemic chemotherapy, and topical pharmacological treatments [6]. The choice of treatment is influenced by various factors, including tumour location, size, histological subtype, the patient’s overall health, and the dermatologist’s experience [7].

Topical pharmacological treatment is a promising approach for localised and early-stage disease owing to its self-administrative and non-invasive nature, the ability to treat a broad area of precancerous lesions, and improved cosmetic outcomes [7]. Moreover, NMSC is more common in rural and regional areas, significantly affecting individuals from lower socioeconomic backgrounds [8]. Therefore, topical treatments could be beneficial for those who may find it challenging to bear the costs associated with travelling to major cities for treatment. However, the efficacy of topical agents is often hindered by poor drug penetration, owing to the barrier function of the stratum corneum (SC) [9]. Furthermore, patient adherence is often low due to the need for prolonged treatment, and the potential for local adverse effects, including inflammation and local skin irritation [10].

In the pursuit of improving the therapeutic outcomes of NMSC, various research groups have attempted innovative solutions to counter these challenges. For example, various nanoparticles (NPs), including transfersomes [11], nanostructured lipid carriers (NLCs) [12], polymeric micelles [13], nanoemulsion [14], and metallic NPs [15], among others, have been developed for the treatment of NMSC. Such nanoparticle-based drug delivery systems often present advantages over conventional therapies in delivering drugs into the deeper skin layers using lower concentrations whilst minimising side effects [16].

From the dosage form perspective, topical patches are a promising drug delivery platform for NMSC, including microneedles [17], polymeric patches [18], and hydrogels [19]. These systems have the potential to enhance drug delivery by improving drug penetration and allowing for a sustained release of the therapeutic agent. They can provide a simpler and more convenient application method, thus potentially improving treatment efficacy and patient adherence [20]. Moreover, NPs incorporated into these patch platforms can potentially enhance the drug stability and penetration; protect it from metabolic degradation; and ensure controlled, targeted delivery to the tumour site [21]. 

In this review, we detail an overview of NMSC, the current landscape of its topical treatment, and the emerging technologies to overcome the limitations of existing therapies. To the best of our knowledge, this work unprecedentedly discusses all the current advancements in innovative topical patches for the treatment of NMSC. In addition to this, the authors present our insights into the key considerations and emerging trends in the development of these advanced topical patches.

## 2. Epidemiology of Non-Melanoma Skin Cancer

NMSC is the most common form of cancer globally, which accounts for the majority of skin cancer cases and represents a significant public health concern [1]. It compasses a group of malignancies that originate from the epidermal layer, excluding melanocytes. The primary forms of NMSC are BCC and SCC, accounting for most cases [1]. BCC is the most common type of skin cancer represents approximately 80% of NMSC, originates from basal cells in the lower part of the epidermis, and typically manifests as pearl-like bumps or pinkish patches of skin (Figure 1). BCC is predominantly found on sun-exposed areas, including the face and neck; however, it can appear anywhere on the body. Although BCC grows slowly and rarely metastasises, it can result in significant local tissue destruction and disfigurement if left untreated [22]. Whilst the superficial and nodular subtypes generally pose a lower risk, other subtypes, such as infiltrating, morphoeic, or sclerosing, are more aggressive and present a significant challenge in diagnosis and treatment [23].

On the other hand, SCC accounts for about 20% of NMSC and arises from the squamous cells that make up most of the upper layers of the epidermis. SCC often presents as a firm, red nodule, or a flat lesion with a crusted, scaly surface. It is also commonly located on sun-exposed areas including the hands, ears, and face [24]. Unlike BCC, SCC has identifiable precursor lesions referred to as actinic keratoses (AK). The progression rate of individual AK to invasive SCC is estimated to be 1–10% over a decade. The presence of AKs serves as a significant indicator of high UV exposure and a higher risk of developing NMSC (Figure 2). AK can occasionally facilitate the early detection and treatment of SCC in situ, preventing metastasis and tissue destruction, as SCC is more destructive than BCC [5].

The prevention and early detection of NMSC are critical in reducing its incidence and impact. Although complex genetic and environmental factors contribute to the pathogenesis of NMSC, prevention strategies aimed at reducing sun exposure have been widely adopted [5]. In addition, public education has been fundamental to improving group-oriented awareness and behavioural modifications [4]. Besides, increasing evidence demonstrated that the use of vitamin D [25] and nicotinamide [26] decreases the risk of NMSC. On the other hand, targeted screening within high-risk groups, diligent self-inspections of the skin, and regular monitoring by medical professionals are instrumental in mitigating both the severity and the fatal consequences of cancer by facilitating the early detection of NMSC [4].

## 3. The Current Treatment Landscape for Non-Melanoma Skin Cancer

The management of NMSC spans a spectrum of modalities, from surgical interventions and radiation therapy to systemic chemotherapy and topical pharmacological treatments [6]. These options are guided by various factors, such as tumour location, size, histological subtype, the patient’s overall health, and the dermatologist’s experience. Surgical options encompass standard excision, curettage and electrodesiccation, and Mohs surgery [27]. Radiation therapy and systemic chemotherapy are typically reserved for more advanced cases or situations where surgery is not feasible [7]. Although surgery is the gold standard for treating NMSC with ill-defined tumour margins, many patients are concerned about its aesthetic impact on visible areas such as the face [28]. Whilst Mohs surgery is highly effective with good cosmetic outcomes, it is costly, time-consuming, and often requires specialists who may not be readily available outside major cities [29]. On the other hand, radiotherapy has disadvantages such as a high treatment frequency and duration of up to 7 weeks and side effects including dermatitis, skin thinning, and hair loss, among others [30].

These limitations highlight the importance of early detection, which allows for localised disease management where topical treatments can play a more substantial role. In this context, topical pharmacological treatments provide several benefits, such as non-invasiveness, self-administration, and improved cosmetic outcomes. These treatments range from traditional cytotoxic agents to novel immunomodulators [7]. 

For example, 5-Fluorouracil (5-FU), a cytotoxic substance is usually employed in the treatment of Bowen’s disease (SCC in situ), superficial BCC, and AK [31]. It works by disrupting DNA synthesis primarily by inhibiting thymidylate synthase. The US Food and Drug Administration (FDA) also approved 5-FU cream as the inaugural topical treatment for superficial BCC following a study that demonstrated a 93% success rate in treating 113 superficial BCC lesions [32]. The effectiveness of the twice-daily 5-FU cream was further validated in a subsequent trial for 12 weeks. This treatment regimen resulted in a 90% histological cure rate and an average clinical cure time of 10.5 weeks [32].

Imiquimod (IMQ) is an example of a modulator of the immune response used for the treatment of superficial BCC, AK, and genital warts [33]. It functions as an agonist for Toll-Like Receptor 7. The initial trials demonstrated a histologic clearance rate of approximately 90% for superficial BCC after being treated with IMQ daily for six weeks [34]. Furthermore, doubling the application frequency improved the response rate but also increased the incidence of local skin reactions [35]. In subsequent double-blind randomised controlled trials, an 82% histologic clearance was observed when the IMQ cream was applied five times weekly [36]. These findings led to the FDA approval of 5% IMQ cream for small superficial BCC that is less than 2 cm in diameter and located on the trunk or extremities [37].

In addition to 5-FU and IMQ creams, diclofenac (3%) gel, ingenol mebutate (0.015%) gel, and tirbanibulin (1%) ointment are other commercially available options for the topical treatment of precancerous lesions, as in AK. However, a detailed discussion of these treatments falls outside the scope of this review. Table 1 summarises the commercially available therapeutic agents with physicochemical properties relevant to skin application.

On the other hand, recent advancements in topical therapies for NMSC also represent the use of several other investigational compounds. The topical application of such active compounds including doxorubicin [38], paclitaxel [39], and carvedilol [11] as well as those with a natural origin including curcumin [40], resveratrol [41] and cannabidiol, among others [12] have also shown promising results against NMSC.

**Table 1 pharmaceutics-15-02577-t001:** Overview of common topical pharmacological treatments for non-melanoma skin cancer and precancerous lesions.

Therapeutic Agent	Dosage Form	Strength	Brand Names	Mode of Action	Common Indications	Limitations	Physico-Chemical Properties	Ref.
5-Fluorouracil (5-FU)	Cream	5%	Efudex^®^ Carac^®^	Interferes with DNA synthesis by blocking thymidylate synthase	Bowen’s disease (SCC in situ); superficial BCC; AK	Skin irritation; photosensitivity	Log P (−0.85); molecular weight (130.078 g/mol); melting point (291.8 °C)	[31,42]
Imiquimod (IMQ)	Cream	5%	Aldara^®^	Induces immune response against cancer cells	Superficial BCC; genital warts; AK	Local skin reactions; psoriasis	Log P (2.6); molecular weight (240.30 g/mol); melting point (295 °C)	[37,43]
Diclofenac sodium	Gel	3%	Solaraze^®^	Inhibits COX-2 enzyme, reducing prostaglandin E2 synthesis	AK	Local skin irritation; digestive adverse events	Log P (4.26); molecular weight (318.13 g/mol); melting point (286 °C)	[44]
Ingenol mebutate	Gel	0.015%	Picato^®^	Induces local lesion cell death; promotes an inflammatory response	AK	Local skin irritation	Log P (3.12); molecular weight (430.5 g/mol); melting point (153.5 °C)	[45]
Tirbanibulin	Ointment	1%	Klisyri^®^	Disrupts microtubules by direct binding to tubulin	AK	Local skin irritation; sun sensitivity	Log P (N/A); molecular weight (g/mol); melting point (N/A)	[46]

## 4. Limitations of Current Topical Treatment for Non-Melanoma Skin Cancer

Current topical treatments for NMSC, although effective in many cases, face substantial limitations. These include local side effects, potential systemic toxicity, variable efficacy, poor skin penetration, dosing inconsistency, and challenges with patient compliance [18,47].

Local skin reactions are a common drawback of most topical therapies, with reactions ranging from mild irritation to severe inflammation, which may often lead to treatment discontinuation. For instance, 5-FU use frequently results in skin irritation, mild erythema, and photosensitivity [32]. Moreover, IMQ has been associated with a range of side effects, predominantly local but occasionally systemic, especially when administered at higher doses. Its local side effects include burning, itching, irritation, and erythema, whilst more uncommonly, systemic side effects including psoriasis and pemphigus have been documented [31,37]. Clinically, the severity of local side effects may often be indicative of treatment efficacy, however, treatment compliance could be significantly compromised in patients with high sensitivity to the reactions. In this regard, strategies including the introduction of new compounds, the reformulation of existing agents, or a more simple treatment regimen may be fundamental to reducing toxicity and, in turn, improving the patient’s adherence to the topical treatment [48].

The efficacy of these treatments can also vary significantly based on factors including the type and stage of NMSC, the location and lesion size, and individual patient factors. The outermost layer of the skin, the SC, poses a barrier, preventing the effective delivery of therapeutic agents to the deeper skin layers where invasive skin cancer grows. Whilst IMQ shows promising results for AK and superficial BCC, its efficacy for other types of NMSC remains unsuccessful or unevaluated [49]. This could be partially attributed to the poor skin penetration of topically applied pharmacotherapy. Despite a high IMQ concentration (5% *w*/*w*) in Aldara^®^ cream, the high solubilising capabilities of the formulation increase the affinity of IMQ for the vehicle and decrease its thermodynamic activity, in turn, preventing the partitioning of IMQ into the skin [13]. This can hinder the treatment of NMSC that extends deeper into the skin, leading to suboptimal treatment outcomes or recurrence of the condition [50].

Dosing inconsistency represents another challenge with semi-solid formulations such as topical creams. Patients may apply too much or too little of the medication, resulting in suboptimal responses and variable clinical outcomes [18]. Similarly, treatment regimens require daily or even twice-daily application over several weeks or months, which can be burdensome for many patients. Additionally, the special handling requirements of some cytotoxic drugs, such as the use of gloves and an applicator, can further deter patients from adhering to the treatment [51].

These limitations underscore the need for innovative topical treatment strategies for NMSC aimed at improving efficacy, reducing side effects, enhancing skin penetration, ensuring consistent dosing, and improving patient compliance. As a result, in the following section, the development of innovative topical patches with promising potential for these challenges will be discussed.

## 5. Advances in Topical Patch Technology for Non-Melanoma Skin Cancer Treatment

The development of innovative topical patches offers a promising advancement in the pursuit of effective, patient-centred, and targeted NMSC treatments. By harnessing novel drug delivery systems, these patches may offer a solution to several challenges associated with conventional topical therapies [52]. 

Innovative topical patches come in various forms, each bringing unique benefits to the patients. For instance, microneedle patches feature minute projections that pierce the skin to allow for the direct and deep delivery of therapeutic agents [52]. On the other hand, polymeric patches employ biocompatible and biodegradable polymers that enable controlled and sustained drug release [53]. Furthermore, hydrogel patches, with their high water content, not only provide comfort and flexibility but also enhance skin hydration, a factor that may promote the better absorption of some drugs due to the swelling of corneocytes [19].

All these patch types share a common advantage where they can improve drug penetration into the skin. Through various mechanisms, including skin hydration and direct skin piercing, these patches can facilitate the delivery of drugs to deeper layers of the skin, addressing one of the significant limitations of traditional topical treatments [51,54]. Moreover, recent advancements in formulation have enabled the integration of NPs within these patch systems. NPs can be engineered to encapsulate and release drugs in a controlled manner whilst their small size allows for better skin penetration, further bolstering the potential of topical patches in NMSC treatment [20].

Furthermore, these advanced patch technologies also hold the potential to significantly improve patient compliance. These patches offer consistent dosing, eliminating the risk of patients applying too little or too much of a product. As demonstrated in the scientific literature, reduced dosing frequencies, enabled by the sustained and controlled drug release properties of patches, such as microneedles and polymeric systems, can simplify treatment regimens, making them less disruptive to patients [18]. Additionally, patches minimise the handling requirements especially when dealing with cytotoxic drugs, which can significantly improve patient adherence [51]. In the following sections, we will discuss novel technologies on topical patches, their potential benefits, limitations, and the in vitro and in vivo evidence supporting their use in NMSC treatment.

### 5.1. Microneedle Array Patches

Microneedles have emerged as a revolutionary method for topical and transdermal drug delivery, overcoming the limitations of traditional topical therapies. Microneedles are minimally invasive as the microscopic needles penetrate the SC without reaching the underlying nerves, thereby delivering medication painlessly and effectively [55]. Over the past decade, extensive research and technological advancements have led to a diverse array of microneedle types and manufacturing methods, covered exhaustively in previous reviews [56,57,58,59].

Whilst the detailed aspects of microneedle technology for drug delivery are beyond the scope of this review, it is essential to recognise their potential roles in NMSC management. As a result, this review will explore the potential benefits and drawbacks of microneedle technology in treating NMSC and how it fits into the broader landscape of innovative topical patches for cancer treatment. 

Firstly, stainless steel microneedles containing 5-Aminolevulinic acid (5-ALA) as a photosensitiser using a micro-precision dip coater were presented [60]. By using a 25% *w/v* 5-ALA solution for five dips, they were able to coat each patch with approximately 350 μg of 5-ALA, achieving a delivery efficiency of around 90%. When compared to its topical cream counterpart (~150 µm), the microneedles achieved much deeper skin penetration (~480 µm). According to its in vivo animal study conducted on female Balb/C mice with A20 cancer cells, the microneedles coated with 5-ALA significantly suppressed the subcutaneous tumour growth by about 57%. Conversely, the topical cream containing 5-ALA (5 mg) failed to suppress the tumour volume, leading to tumour growth like the untreated control group.

A study introduced the fabrication of microneedles using poly(L-lactide) that are coated with infrared-responsive PEGylated gold nanorods (GNR-PEG@MN) [61]. Additionally, docetaxel-loaded micelles (MPEG-PDLLA-DTX micelles) were independently synthesised and administered to female A431 tumour-bearing Balb/cA nude mice to evaluate their combined effects with GNR-PEG@MN. The GNR-PEG@MN, with a height of 480 μm, demonstrated excellent skin penetration capabilities and posed no harm to the skin whilst achieving effective heat transfer in vivo, with the tumour sites reaching 50 °C within 5 min. When compared to standalone chemotherapy and photothermal therapy, the combination of low-dose MPEG-PDLLA-DTX micelles and GNR-PEG@MNs eliminated the A431 tumour in vivo with no recurrence, showcasing a significant synergistic effect. Therefore, GNR-PEG@MN could potentially serve as an effective carrier to boost the anti-tumour impact of MPEG-PDLLA-DTX micelles for the treatment of superficial tumours.

Another study presented the development of a hyaluronic acid dissolvable microneedle patch using near-infrared light-responsive monomethoxy-poly(ethylene glycol)-polycaprolactone NPs containing 5-FU and indocyanine green (5-Fu-ICGMPEG-PCL@HA MN) [17]. The microneedle system had a good skin penetration of 600 µm with a rapid heating transfer efficacy to 60 °C in 5 min upon 808 nm near-infrared laser. Moreover, in vivo studies using A431 tumour-bearing Balb/cA nude mice demonstrated the tumour inhibition capability of 5-Fu-ICG-MPEGPCL@ HA MN with no recurrence, which suggests the synergistic effect of photothermal therapy and chemotherapy.

The microneedle patch platform was also fabricated using biocompatible photopolymer resin via stereolithography 3D printing [62]. The microneedles were subsequently coated with cisplatin formulations using an inkjet dispensing method (Figure 3). The study showed that the 3D-printed microneedles demonstrated excellent skin penetration, achieving 80% penetration depth (737.7 ± 63.7 µm). In vitro release studies revealed that cisplatin was released rapidly, with 80–90% of its payload released within the first hour. In vivo testing on A431 human squamous carcinoma xenografts in BALB/c nude mice demonstrated that the cisplatin permeated sufficiently, exhibiting high anticancer activity, and resulting in 100% tumour regression. The histopathological analysis also demonstrated the tumour inhibitory effect, with the presence of clearly defined lesions with thin fibrous capsules and necrotic cores. The study also highlighted the advantages of 3D printing technology in fabricating microneedles, such as cost-effectiveness, accuracy, reproducibility, and the potential for upscaling.

A recent study reported the development of microneedles loaded with IMQ, utilising a polyvinylpyrrolidone-co-vinyl acetate polymer [63]. The microneedle patch demonstrated a penetration depth of 426 ± 72 µm in the porcine skin model. The ex vivo permeation studies revealed that despite the microneedle containing an IMQ load six times lower than the clinically relevant dose of Aldara^®^, which is commonly used for BCC treatment, it achieved a similar level of IMQ intradermal delivery. Furthermore, the time-of-flight secondary ion mass spectrometry analysis of skin cross-sections showed that IMQ was localised within the skin following delivery via the microneedle, whereas skin treated with Aldara^®^ displayed the drug predominantly within the SC. Table 2 summarises the developed microneedle patches for the potential treatment of NMSC.

### 5.2. Polymeric, Drug-in-Adhesive, and Matrix-Type Patches

Since their first introduction in the 1980s, topical patches have evolved significantly with a variety of types, such as polymeric, reservoir, matrix, drug-in-adhesive (DIA), microneedles, and smart skin-adhesive patches. Polymeric patches are a broad term that includes reservoir, matrix-type, and DIA patches, utilising polymers to form the structure and achieve a wide range of properties [64]. The history, types, preparation, and materials of these patches have been extensively reviewed through a large body of literature precedence [64,65,66,67]. Briefly, reservoir patches utilise a drug-storing reservoir placed between a backing layer and rate-controlling membrane, whereas the drug is directly incorporated into the polymeric matrix or adhesive layer in the matrix-type and DIA patches (Figure 4). In this review, our focus lies on the most prevalent types—DIA and matrix-type patches—within the context of NMSC treatment. Unlike reservoir patches, DIA and matrix-type patches can be tailored and cut to any size and shape, making them potentially more suitable for addressing the heterogeneity of skin cancer.

Firstly, a study reported the development of polymeric patches with varying concentrations of IMQ at 4.75, 9.50, and 12.50 mg/cm^2^ [18]. The patches exhibited a bioadhesion of approximately 1.76 N/cm^2^ when removed from neonate porcine skin, indicating a strong adherence of topical application to the skin. Interestingly, these patches demonstrated a significantly higher drug release across a Cuprophan^®^ dialysis membrane (10,500 DA cut-off) compared to the commercially available cream, Aldara^®^, over 6 h. This suggests that the polymeric patches could potentially offer a more efficient delivery of IMQ, thereby enhancing its therapeutic efficacy.

In another study, the development of polymeric film containing gold nanorods (GnRs) for use in local hyperthermia applications was reported [68]. The GnRs functionalised with thiolated poly(ethylene) glycol to improve biocompatibility were incorporated into a crosslinked polymeric film made of copolymer of methyl vinyl ether and maleic acid (Mw = 1,200,000). The authors noted that the GnRs remained entrapped within the polymeric network even after the film swelled. The films did not leave any polymeric residues on the porcine skin, demonstrating their improved biocompatibility. Furthermore, the GnR-loaded films were capable of heating the skin model to over 40 °C, validating their potential for non-invasive local hyperthermia treatments against NMSC.

Moreover, DIA patches containing 5-FU were developed by our research group for the first time, using a cationic copolymer consisting of dimethylaminoethyl methacrylate, butyl methacrylate and methyl methacrylate (2:1:1) (Eudragit^®^ E) as an adhesive polymer matrix [51]. As adhesion is one of the critical parameters for topical application, various plasticisers were screened and optimised by adjusting the plasticiser-to-polymer ratio to achieve the best adhesive properties of the patches. The authors suggested that the patches containing 40% (relative to the polymer ratio) triethyl citrate, dibutyl sebacate, or triacetin as a plasticiser achieved adhesive properties similar to those of several marketed patches. This study also demonstrated a controlled release of 5-FU from the patches, suggesting its potential application in skin cancer treatment.

Continuing this success, another DIA patch using Eudragit^®^ E and triacetin was also developed by our group. In this patch, a combination of both 5-FU and IMQ was evaluated for their potential use in NMSC (Figure 5) [69]. The patches were formulated to contain 81.4 ± 0.57 μg/0.64 cm^2^ of 5-FU and 82.5 ± 0.50 μg/0.64 cm^2^ of IMQ. The release rate of 5-FU was observed to be faster than that of IMQ in vitro, with about 75% of the drug content being released within 50 min and 120 min, respectively. Both 5-FU and IMQ demonstrated an initial burst release, which may be advantageous for topical applications. This release pattern facilitates the rapid establishment of a high concentration gradient, thereby enhancing the diffusion force across the skin to increase permeation for the initial application period. The authors of the study also highlighted the need for further studies to assess the efficacy and safety of the patches in skin cancer models.

In a recent study, NLCs containing IMQ were developed by using Design of Experiments [20]. The NLCs were developed using stearyl alcohol, oleic acid, polysorbate 80 and stearoyl polyoxyl-32 glycerides. The optimised formulation was further formulated into a matrix-type topical patch consisting of hydroxypropyl methylcellulose (HPMC) K4M and propylene glycol. The ex vivo deposition study demonstrated that the IMQ-NLCs patch significantly increased IMQ deposition in the deeper skin layers than the commercial cream. Specifically, the patch deposited 3.3 ± 0.9 μg/cm^2^ into the dermis layer and 12.3 ± 2.2 μg/cm^2^ into the receptor chamber, whilst the commercial cream resulted in the deposition of 1.0 ± 0.8 μg/cm^2^ and 1.5 ± 0.5 μg/cm^2^ of IMQ, respectively (Figure 6). The authors concluded that IMQ-NLC-loaded patches hold great potential as a topical treatment strategy for skin cancer with improved drug delivery and patient adherence. They also highlighted the possibility of adopting environmentally friendly practices in the development of NLCs by reducing energy and solvent consumption. The examples of developed DIA and matrix-type patches with potential applications in NMSC are summarised in Table 3.

### 5.3. Hydrogels

Hydrogels are typically made of hydrophilic polymers and feature a unique cross-linked three-dimensional network capable of absorbing substantial quantities of water [70]. Generally, hydrogels originate from various sources including natural materials, such as chitosan, hyaluronic acid and alginate, among others. They may also have a synthetic origin, such as polyethylene glycol, polyvinyl alcohol, sodium polyacrylate, and their related copolymers, as well as a hybrid nature, combining materials from both origins [19]. Hydrogels have served as a versatile drug delivery platform due to their customisable physical properties, controllable degradation rate, and various drug-encapsulation capabilities [71]. Their specific applications in skin cancer therapies have previously been summarised in a recent review [19]. 

Within the context of NMSC, doxorubicin-loaded hydrogels were developed using natural polymers, including dextran, chitosan, gelatine, and xanthan cross-linked with acrylamide and N,N′-methylenebis(acrylamide) for topical application [72]. The developed hydrogels demonstrated swelling and bioadhesion in simulated biological fluid and membrane. The authors also highlighted that doxorubicin was released from the hydrogels over a minimum of 50 h and was highly effective against the A431 epidermal cell line in vitro, whereby the sustained release of doxorubicin interrupted cell division and induced cell apoptosis.

## 6. Challenges Associated with Novel Topical Patch Development for Non-Melanoma Skin Cancer Treatment

Whilst the development of innovative topical patches for NMSC treatment is promising, it also faces many challenges. NMSC often presents as plaques, nodules, or lesions with a thickened or hyperkeratotic SC. This thickened, scaly SC can pose a significant barrier to the penetration of topical formulations and increase the path length for passive diffusion of active ingredients, potentially diminishing the effectiveness of treatments [16]. Microneedles, for example, can directly deliver the drugs to the tumour site by piercing through the SC, although they often face several formulation and technical challenges. Some of the challenges may include limited overall drug loading capacity, the poor loading of hydrophobic drugs, dosing inaccuracy, and achieving sustained release, among others [73,74].

Furthermore, various nano-formulations have also been developed to potentially improve skin penetration. However, the biological or clinical responses at the cellular level after exposure to the skin tumour microenvironment can present further challenges [75]. Factors such as skin and tumour permeability, anatomical site, skin hydration, pH, hair follicle density, sebum production, and individual responses can also influence the ability of NPs to accumulate within the tumour. Additionally, the payload must overcome challenges related to its own physical and chemical characteristics, such as molecular weight, hydrophilicity, and charge, to be absorbed by the target cells and induce therapeutic actions [76].

Moreover, most studies evaluating novel topical therapy have been conducted on in vitro cell lines or in vivo mouse models of NMSC. However, there are significant differences between human and mouse skin, such as the origin of neoplasms from a molecular, anatomical and physiological perspective [77]. For example, the skin of a mouse is looser, has a higher density of hair follicles, and has a very thin epidermis compared to human skin. This difference extends to interactions between the tumour cells and their surrounding epidermal–dermal environment. Therefore, the findings from studies utilising mouse models may not directly translate to the human response to treatments. For a more accurate representation of the tumour and human skin, evaluating topical formulations on bioengineered human skin equivalents of NMSC or live patient skin explants would be beneficial [78].

Patient compliance is another significant challenge. Whilst patches offer the advantage of localised treatment with potentially fewer systemic side effects, some patients may find certain patch types inconvenient or uncomfortable to use. For example, microneedle patches, whilst potentially more effective at delivering drugs through the skin, may still be perceived as invasive by some patients [79]. Furthermore, topical patches need to be formulated to provide adequate adhesion to stay in place for the required duration whilst they should not cause discomfort or damage when removed. In addition to this, the patch must be able to withstand the effects of sweat and other skin secretions, as well as the mechanical forces of skin movement [66]. 

## 7. Key Considerations for Novel Patch Development

The development of novel topical patches for NMSC treatment is an intricate process that requires the careful balance of many factors. These patches must be designed not only to effectively deliver therapeutic agents to the target site but also to ensure the patient’s safety, comfort and compliance. The following key considerations are crucial in the development process, and addressing these can significantly enhance the potential of topical patches as a promising strategy for NMSC treatment. Whilst various considerations for developing microneedles to bypass the skin barrier by directly piercing the SC have been extensively discussed previously in the literature [73,80,81,82], our discussion will mainly focus on other types of topical patches, including DIA and matrix-type patches.

### 7.1. Physicochemical Properties of Drugs

The intrinsic physicochemical properties of a drug play a critical role in drug permeation through the skin. Broadly speaking, soluble compounds (usually indicated by a low melting point) with a small molecular weight (<about 500 g/mol) that possess moderate lipophilicity (with a logarithm of the octanol-water partition coefficient in the range of 1 to 3, Log P) and few hydrogen bonds may penetrate through the SC [45,83]. Based on these criteria, for instance, 5-FU with a molecular weight of 130.078 g/mol and Log P of −0.85 may not readily penetrate across the SC without the aid of permeation-enhancing techniques due to relatively high hydrophilicity. To aid the selection of a drug candidate for potential topical applications, an extensive list of drugs with these parameters relevant to skin permeation has been presented in a recent article [45].

### 7.2. Type of Patches and Polymer Selection

As previously discussed on different types of patches from the reservoir to DIA patches (Section 5), selecting a patch type may largely depend on the specific therapeutic needs of the patient, physicochemical properties of a drug candidate, the desired release profile, and the ease of manufacturing, among others. For instance, reservoir patches can offer much tighter drug release; however, they pose a risk of burst release and drug leak [84]. On the other hand, DIA patches have the advantages of simplicity and flexibility. For matrix and DIA patch development, the selection of matrix-forming polymers and pressure-sensitive adhesives (PSA) is one of the most important considerations to ensure adequate drug solubility as well as achieving optimal mechanical and adhesive properties of the patches [85]. 

Commonly used PSAs are acrylics, polyisobutylene (PIB), and polysiloxane, many of which are commercially available in different grades [66]. Acrylic-based PSAs are synthesised by adding hard and soft monomers in varying ratios, allowing for the customisation of the final characteristics. Acrylic-based PSAs generally exhibit greater oxidation resistance compared to PIB-PSAs owing to their saturated functional groups. Additionally, they are colourless and transparent, and maintain their colour stability even when exposed to sunlight, i.e., not turning yellow over time [66]. 

On the other hand, PIB-PSAs can be formulated by either combining PIBs of high and medium molecular weights or by incorporating low-molecular-weight polybutylene into this mix. The first method results in lower peel adhesion values, which further decrease as the ratio of medium-molecular-weight PIB increases. The second method, which includes low-molecular-weight polybutylene, provides a wider variety of PIB blends and enhances the adhesive characteristics of the matrix in terms of tack and peel adhesion. However, the use of PIBs comes with certain drawbacks, including their susceptibility to oxidation and low permeability to air and water vapour [86]. 

Silicone PSAs consist of a long-chain polymer, specifically polydimethyl siloxane (PDMS), and a benzene-soluble silicate resin. The resin exhibits a high glass transition temperature (T_g_), whilst the polymer has a notably low T_g_, around −140 °C. The raw material is supplied as a blend of these two components, typically in heptane. The ratio of resin to PDMS determines the characteristics of the final product. Generally, a higher ratio of resin results in a harder final silicone [86]. Whilst silicon-based PSAs are known for their excellent drug diffusivity, they also have a pronounced tendency towards drug crystallisation [66,87].

### 7.3. Prevention of Drug Crystallisation

Drug crystallisation is often more prevalent in matrix systems, as both the drug and excipient could experience phase changes over time. For instance, a dissolved drug may crystallise, or a dispersed drug may agglomerate; these instabilities could negatively impact the adhesion and drug release of the system. Therefore, the initial screening study, which includes solubility measurements and dispersion stabilisation, is particularly crucial for matrix and DIA systems [84]. A simple and commonly used method is slide crystallisation studies [88,89,90].

Briefly, the saturation solubility of a drug in various adhesives can be determined by preparing a drug with a series of concentrations mixed with adhesives. Then, a smear of the mixture on a glass slide is observed under the optical microscope for the presence of drug crystals. In a recent study [90], the saturation solubility of 4-benzylpiperidine was evaluated in different adhesives where its maximum solubility was determined to be less than 4.5% in silicone (BIO PSA^®^ 7-4301) and PIB (DURO-TAK^®^ 87-6908) adhesives (Figure 7).

Despite the best efforts in adhesive screening, drug crystallisation can still be an issue in many cases. One of the simplest and most effective methods may be incorporating crystallisation inhibitors into the formulation. Some of the examples include polyvinylpyrrolidone, along with its derivatives; copovidone; crospovidone; mannitol; polyethylene glycol; isopropyl myristate; dextrin derivatives; polypropylene glycol; polysorbate 80; poloxamer; glycerine; and caprylocaproyl polyoxyl-8 glycerides [91]. In a study, the addition of isopropyl myristate (10% *w*/*w*) into a BIO PSA^®^ 7-4301 formulation helped to incorporate a higher concentration of 4-benzylpiperidine (up to 10% *w*/*w*) without experiencing separation compared to its counterpart with no crystallisation inhibitor (<4.5% *w*/*w*) [90].

Furthermore, encapsulating drugs into NPs has also been explored as a potential recrystallisation-inhibition approach in patch formulations [92]. In the study, ibuprofen and hydrocortisone as model drugs were incorporated into several different nanosystems, including nanoemulsions, solid lipid nanoparticles (SLNs), and polymeric NPs, and further formulated into HPMC K100M patches. The results exhibited that the control patches loaded with the drugs in their free form exhibited the most significant crystallisation whilst the patches formulated with nanoemulsion and SLNs demonstrated the least degree of crystallisation. Furthermore, this decrease in crystallisation also enhanced drug permeation through the skin ex vivo, highlighting the function of lipid and polymeric NPs as crystallisation inhibitors and permeation enhancers [92].

### 7.4. Backing Layer and Release Liner Selection

In the matrix and DIA system, the adhesive matrix is placed between the release liner and the backing layer, each serving a distinct purpose in its design and function (Figure 4). The backing layer is the outermost part of the patch that comes into contact with the environment. It is designed to protect the patch from external factors and to provide structural support. The backing layer must be flexible to allow the patch to conform to the skin’s surface and movements [90]. A number of materials can be used for the backing layer, including polyester, polyethylene or polyurethane, among others [51].

On the other hand, the release liner is a protective layer that covers the adhesive side of the patch during storage and before application. It is removed just before the patch is applied to the skin. The release liner must be easy to peel off without affecting the integrity of the adhesive layer and the overall structure of patches [93].

The section process of the backing layer and release liner can be guided by their affinity to the adhesive layer [90,94]. The backing layer is required to always maintain a strong affinity with the adhesive layer. This bond can be tested by applying a drop of the formulation onto the backing layer, followed by drying in a fume hood to allow the adhesive solvent to evaporate. The adhesiveness and peel strength of the formulations on individual backing layers can then be evaluated using a gloved hand. A backing layer with the strongest affinity to the adhesive is generally recommended. A similar approach can be used for selecting a release liner; however, the one with the least affinity to the adhesive is preferred as the release liner as it should be removed easily prior to its skin application. According to a recent study [90], PIB-based PSAs showed the highest affinity to polyethylene backing layer, whilst the release liner made of polyester was the preferred choice.

### 7.5. Mechanical and Adhesive Properties

The mechanical and adhesive properties of a patch play a pivotal role in its overall performance. Optimal flexibility and mechanical properties allow the patches to conform to skin folds and creases without breaking and compromising their adhesion or causing discomfort. On the other hand, adhesion ensures that the patch maintains full contact with the skin throughout its application period. When the patch adheres to the skin, the hydration of the patch facilitates the partitioning of the drug between the patch and the skin, serving as the primary mechanism for drug delivery into or through the skin [90]. 

Several studies discussed in vitro methods for evaluating the mechanical and adhesive properties of patches using a texture analyser [51,88,90,93]. The authors suggest that the mechanical properties may be assessed based on the tensile strength and elongation at break (%), which can be adjusted by changing the plasticiser-to-polymer ratio. According to Kim et al. [51], the tensile strength of Eudragit^®^ E patches has an inverse relationship with the concentration of plasticisers such as triacetin, dibutyl sebacate and triethyl citrate, whilst elongation at break (%) showed a direct relationship with them.

Moreover, adhesion can be measured in vitro using tack (initial adhesion), shear strength (resistance to slippage during wear), and peel adhesion (force required to remove the adhesive from the skin) [93]. Briefly, for the measurement of tack, a stainless probe is allowed to have short contact with the adhesive side of the patches and the force at which the detachment occurs is measured. On the other hand, peel adhesion can be evaluated by measuring the force required to remove the patches from flat surfaces, such as stainless steel or Teflon^®^ surfaces. Whilst the patches should remain intact during and after the peel studies, a recent study [51] described various adhesive and cohesive failures that can happen during the testing (Figure 8).

### 7.6. In Vitro Drug Release Profile

In the development of topical patches for drug delivery, achieving the desired drug release profile is a critical aspect. This is because the release profile determines the rate at which the drug is delivered to the target site, which in turn affects the efficacy of the treatment. Various testing methods, such as the USP method 5 Paddles over Disk and Franz diffusion cell apparatus are used to evaluate the drug release profile of topical patches [89,95].

Considering NMSC often manifests as hyperkeratotic or SC-thickened plaques, lesions or nodules, it might be advantageous to have a bi-phasic release profile exhibiting an initial burst release pattern followed by a slow release [16,96]. The initial burst release may be helpful to set up a high concentration gradient, thereby augmenting the diffusion force across the skin and facilitating rapid permeation for the initial period to achieve immediate effects [97,98]. The subsequent sustained release maintains a steady concentration of the drug at the target site, ensuring long-term effectiveness. Several topical formulations have shown such a release pattern and demonstrated effectiveness against various skin cancer models [96,99,100]

Depending on the desired target release profile, different formulation approaches can be used to adjust the release profile of topical patches, including the incorporation of polymers with different hydrophilicity or the use of NPs. For instance, olanzapine-loaded patches containing different ratios of Eduragit^®^ RL (more hydrophilic) and RS (more hydrophobic) have been developed [101]. The authors claimed that the RL and RS ratio of 3:2 achieved the release and permeation profile of olanzapine as a patch system that can potentially be used for up to 72 h. Moreover, curcumin-loaded SLNs were fabricated and incorporated into a patch based on polyvinyl alcohol [102]. Similarly, the patch with SLNs released the curcumin much longer for up to 72 h, whilst the patch without SLNs was much short-lived.

### 7.7. In Vitro and Ex Vivo Permeation Profile

Elucidating drug permeation profile is another critical aspect of topical formulation development. Franz diffusion cells have been widely used as a reliable method for evaluating drug permeation across the skin. Whilst excised human skin remains the gold standard, although its cost, ethical considerations, and inherent variability, amongst others, have limited its use [103]. Although many alternative animal skins are available, including pig, rat, mouse, rabbit, and snake, significant differences exist between animal and human skin, from an anatomical, physiological, and molecular perspective to interactions between the tumour cells and its surrounding epidermal–dermal environment [77]. Therefore, these differences need to be taken into account during the establishment of ex vivo models using animal skins. Nonetheless, pig skin has been extensively used in research and pharmaceutical development due to its histological and physiological similarities to human skin [104]. However, the use of biological membranes can pose challenges, including intricate preparation procedures, storage constraints, and batch-to-batch variability, frequently leading to variability in results [105]. To address these issues, synthetic membranes, such as Strat-M^®^, have recently been used and demonstrated a good correlation with human skin, offering a more standardised and accessible alternative for the permeability evaluation of several topical products during formulation development and optimisation [106,107]. Whilst synthetic membranes be a valuable tool as an initial screening tool, they cannot fully replicate the complex heterogeneity of human skin, including aspects of cell metabolism and skin appendages. Therefore, findings should be complemented with data obtained from human skin studies or in vivo results to provide a more comprehensive and accurate data interpretation [103].

On the other hand, given that invasive NMSC can originate from the squamous or basal cells of the epidermis and potentially spread to deeper tissues, deposition studies evaluating drug content in different skin layers are invaluable [20,63]. Generally, tape-stripping techniques are employed to provide insights into the distribution of drugs within the skin and their potential to reach the target site of action [108]. The tape-stripping method involves the sequential removal of the SC using adhesive tape (commonly up to 20 times), followed by the extraction and quantification of the drug from each tape strip. Additionally, the heat separation method is used to separate the epidermis from the rest of the skin, further facilitating the analysis of drug distribution [109]. This approach offers a detailed profile of drug distribution within the skin, thereby ensuring the delivery of the therapeutic agent to the target site.

Considering drug penetration into deeper skin layers including the dermis and subcutaneous tissue is desired to target invasive NMSC, various permeation-enhancing technologies have been developed, including the use of chemical permeation enhancers. Briefly, chemical permeation enhancers including fatty acids, alcohols, esters, amines, and surfactants, amongst others function by disrupting the lipid structure in the SC, thereby facilitating passive drug diffusion through the skin [110]. Chemical permeation enhancers bring several benefits, including versatility, a commendable safety profile, ease of incorporation into formulations, and cost-effectiveness. Furthermore, their compatibility with simplified scale-up processes enhances their accessibility, making them a popular choice for widespread application in topical drug delivery [111]. For instance, a few studies have demonstrated the efficacy of various chemical permeation enhancers in facilitating the delivery of 5-FU through excised human skin. One study demonstrated that skin samples pretreated with oleic acid; 1,8-cineole; menthone; and nerolidol enhanced the permeation of a 5-FU solution up to 24-, 95-, 42-, and 25-fold, respectively [112]. Similarly, another study revealed that aqueous solutions of 5-FU, when combined with 5% (*w*/*v*) isopropyl myristate, 5% (*w*/*v*) lauryl alcohol, or 3% (*w*/*v*) Azone^®^, resulted in an up to 3-, 4-, and 24-fold improvement in 5-FU permeation through excised human SC [113]. Nonetheless, detailed reviews on the chemical permeation enhancers for topical drug delivery are available in several previous articles [111,114,115].

Nanotechnology represents another promising approach to enhancing skin permeation. Various nanocarriers, such as liposomes [116], niosomes [117], SLNs [118], NLCs [41], polymeric NPs [100], nanoemulsions [119], and metallic NPs [120], amongst others, have been explored for their potential to improve drug delivery to the skin cancer. Briefly, these nanocarriers can enhance drug stability, prolong release, increase skin penetration, and allow for targeted delivery, although these may be associated with several challenges, including stability issues, potential toxicity, and the complexity of formulation and scale-up processes [121,122]. Comprehensive discussions detailing the advantages and disadvantages of each nano system relevant to skin cancer can be found in previous publications [16,123,124,125].

In the context of treating more invasive forms of NMSC, achieving drug penetration into deeper skin layers, including the dermis layer, is deemed to be crucial whilst it could also potentially increase the risk of systemic exposure. In a few studies, superior cytotoxicity against in vivo mouse skin cancer models has been demonstrated using optimised formulations that achieved the highest skin permeation ex vivo [126,127]. However, the correlation between the concentration of a topically applied drug in the skin as determined ex vivo, therapeutic action at the site of action, and potential systemic exposure remains largely unexplored [128]. This underscores a significant gap in our understanding of topical drug delivery for NMSC treatment, emphasising the need for establishing the efficacy and safety of these therapies, determined by drug biodistribution and pharmacokinetics.

### 7.8. Stability

The stability of the patch is crucial, including both the drug and the patch components, in which the patch should maintain its integrity, whilst the drug should remain potent over the intended shelf-life. In the early 2000s, the stability framework was established by the International Conference on Harmonization of Technical Requirements of Pharmaceuticals for Human Use in collaboration with the World Health Organization [129].

Generally, stability is determined by five key aspects, namely chemical, physical, therapeutic, microbiological and toxicological stability [130]. Within the context of patches, the European Medicines Agency (EMA) recommends the careful monitoring of alterations in the thermodynamic activity of the drug substance, such as crystallisation and changes in the excipient behaviours. They also suggest performance tests, including in vitro release, skin permeation, as well as adhesive properties, over the intended period of storage [131].

The parameters for stability testing, including storage conditions, testing frequency, the number of replicates, and the duration of studies, should be clearly defined in the dossier to account for usage, shipment, and storage periods. Long-term testing should be conducted for at least 12 months at either 25 °C ± 2 °C/60% relative humidity (RH) ± 5% RH or 30 °C ± 2 °C/65% RH ± 5% RH. Moreover, both intermediate and accelerated testing should cover at least 6 months at conditions of 30 °C ± 2 °C/65% RH ± 5% RH and 40 °C ± 2 °C/75% RH ± 5% RH, respectively [130].

### 7.9. Other Considerations

In addition to the technical aspects of patch development, other critical considerations can significantly impact the success of a topical patch product. The design of the patch and the choice of materials should not only meet the therapeutic requirements but also facilitate an efficient, reproducible, and scalable manufacturing process. This includes considerations for the ease of raw material sourcing, the complexity of the manufacturing process, and the ability to maintain consistent quality in large-scale production. Regulatory considerations are paramount in the development of a topical patch. The patch must meet the safety and efficacy requirements of regulatory bodies such as the FDA or EMA to demonstrate the quality, safety, and efficacy of the patch [131].

Lastly, in the era of increasing environmental consciousness, the principles of green chemistry and sustainability are becoming increasingly important in the field of pharmaceutical development [132,133]. The selection of environmentally friendly materials, the minimisation of waste, and the use of energy-efficient processes are all aspects that need to be considered. Moreover, the concept of sustainability goes beyond green chemistry, which utilises a holistic approach based on three themes including environment, economy, and society [134]. Whilst pharmaceutical development should not compromise environmental and human health, our approaches should also prioritise cost-effectiveness and accessibility for a broader population in need. Furthermore, new pharmaceutical developments should aim to address current practical and industrial challenges that can yield long-term benefits for both society and industry [135].

## 8. Future Perspective

The development of topical patches for NMSC treatment presents a promising avenue for enhancing treatment outcomes and patient compliance. The advancements in nanotechnology, polymer science, and drug delivery systems have opened new possibilities for the design of patches that can effectively deliver therapeutic agents to the target site. As we move forward, it is crucial to continue research and innovation in this field. Here are a few potential recommendations for future directions.

The use of combination therapy, in which two or more therapeutic agents are used together, could be a promising strategy for increasing the effectiveness of topical patches for NMSC. This approach could involve the use of multiple chemotherapeutic drugs with distinct modes of action. The rationale behind combination therapy is that it can potentially achieve synergistic effects and, in turn, enhanced efficacy. For instance, a recent study combining 5-FU and CBD formulated in a nano-formulation has shown synergistic effects in vitro against the human epidermoid carcinoma cell line [12]. Moreover, the improved efficacy of the co-administration of a 5-FU and IMQ cream was demonstrated in patients with SCC in situ after 6 weeks of treatment [136]. However, designing such combination therapy patches would require careful consideration of factors like the compatibility and stability of the drugs, and the potential for drug-drug interactions.

Furthermore, topical patches could potentially be designed to deliver novel immunotherapeutic agents that stimulate the body’s immune system to fight against more aggressive NMSC including metastasis. These patches could contain agents like immune checkpoint inhibitors, cytokines, or even genetically modified cells that can activate immune responses against cancer cells. For instance, patches could be designed to deliver agents like PD-1/PD-L1 inhibitors, which have shown promise in treating various types of skin cancer, including melanoma metastasis, although this would present significant technical and formulation challenges [137]. Therefore, this approach would require careful design to ensure the stability and efficacy of the immunotherapeutic agents, as well as to control their release and permeation into the skin.

Moreover, the heterogeneity of skin cancer presents a unique challenge in treatment, as the disease varies greatly in its manifestation from patient to patient. The advent of 3D printing technology in pharmaceuticals could play a significant role in this personalisation where personalised topical patches for skin cancer treatment can be created on demand [138]. These patches can be designed with controlled network topology and loaded with specific therapeutic agents, providing a highly customisable approach to treatment with their design, size, and dimensions. For instance, the application of 3D printing technologies in the fabrication of topical patches has been demonstrated in the field of wound healing [139]. This innovative approach could be similarly employed to create personalised topical patches for skin cancer treatment, offering a promising strategy to address the heterogeneity of skin cancer including the shape, size, and depth of invasion. However, the practical implementation of this approach would require overcoming significant regulatory and technical challenges.

## 9. Conclusions

In conclusion, topical patches hold significant potential for NMSC treatment. Innovative approaches such as combination therapy and personalised patches could transform the landscape of NMSC management. However, technical and regulatory issues, patient compliance, and proving in vivo efficacy remain challenging, and overcoming such challenges will require collaborative efforts from researchers, clinicians, and industry. Despite these hurdles, the future of topical patches in NMSC treatment is promising, paving the way towards effective, personalised, and minimally invasive treatment strategies.

## Figures and Tables

**Figure 1 pharmaceutics-15-02577-f001:**
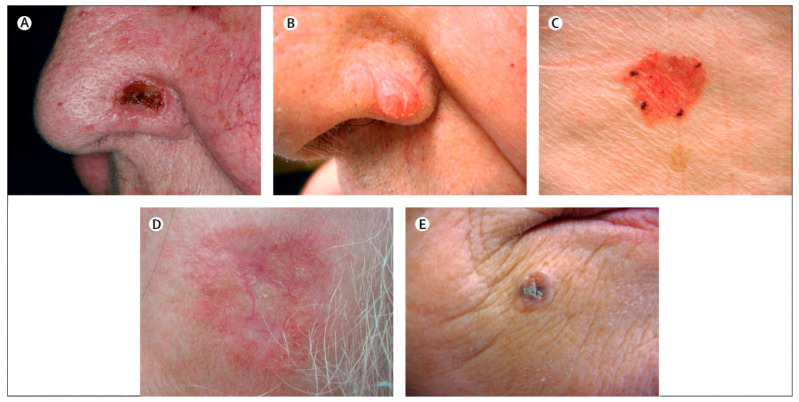
Clinical subtypes of basal-cell carcinoma including (**A**) classic rodent ulcer, (**B**) cystic or nodular, (**C**) superficial, (**D**) morphoeic, and (**E**) pigmented basal cell carcinoma. Reprint with permission from Madan et al. [5], Elsevier, 2023.

**Figure 2 pharmaceutics-15-02577-f002:**
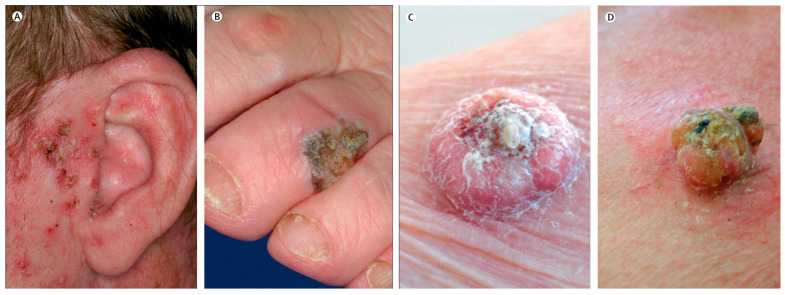
Clinical presentations of precancerous lesions and squamous-cell carcinoma, including (**A**) actinic keratoses, (**B**) squamous cell carcinoma in situ (Bowen’s disease), (**C**) keratoacanthoma and (**D**) squamous cell carcinoma. Reprint with permission from Madan et al. [5], Elsevier, 2023.

**Figure 3 pharmaceutics-15-02577-f003:**
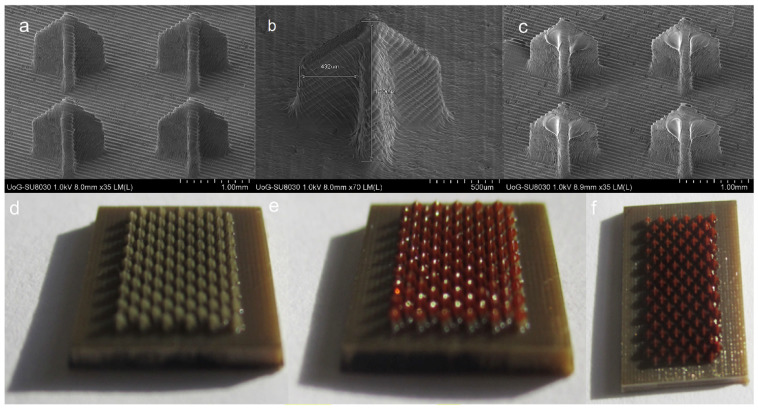
SEM images of (**a**,**b**) uncoated and (**c**) coated 3D-printed cross–MNs. Optical images of (**d**) uncoated and (**e**,**f**) coated with red dye 3D printed cross–MN patches. Reprint with permission from Uddin et al. [62], Elsevier, 2023.

**Figure 4 pharmaceutics-15-02577-f004:**
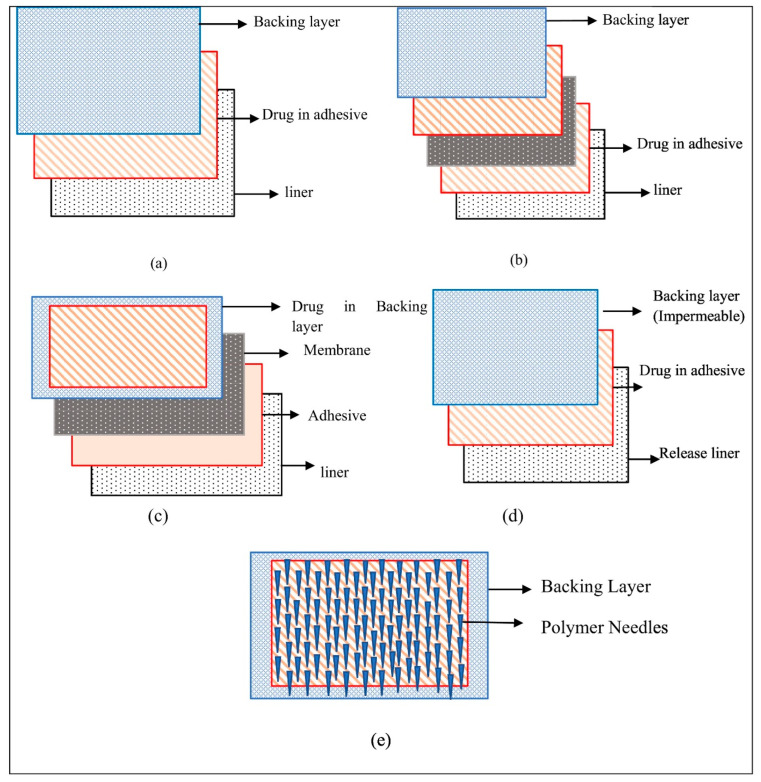
Different types of patches. (**a**) Single-layer drug-in-adhesive, (**b**) multi-layer drug-in-adhesive, (**c**) reservoir transdermal patch, (**d**) matrix patch, and (**e**) microfabricated microneedles. Reprint with permission from Sabbagh et al. [64], Elsevier, 2023.

**Figure 5 pharmaceutics-15-02577-f005:**
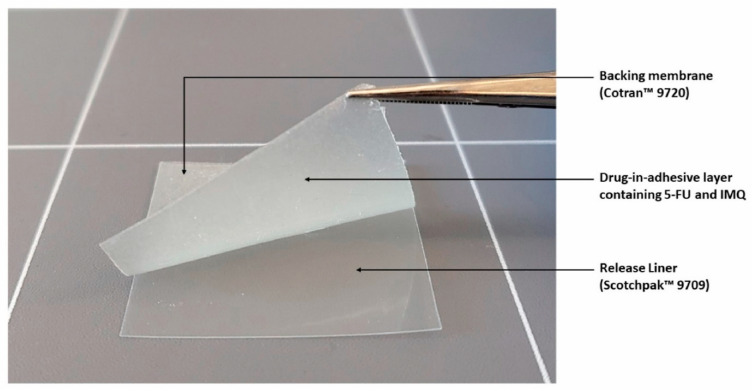
A drug-in-adhesive patch loaded with 5-FU and IMQ. Reprint with permission from Kim et al. [69], Elsevier, 2023.

**Figure 6 pharmaceutics-15-02577-f006:**
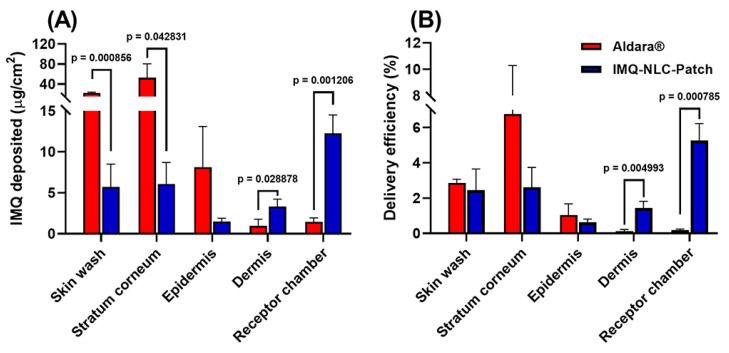
(**A**) IMQ skin deposition and permeation ex vivo after 24 h from the IMQ-NLC-Patches. (**B**) The delivery efficiency from IMQ-NLC-Patch represented as a percentage of the applied dose. *p*-Value is shown when a statistically significant difference is achieved. Reprint with permission from Kim et al. [20], Dove Press, 2023.

**Figure 7 pharmaceutics-15-02577-f007:**
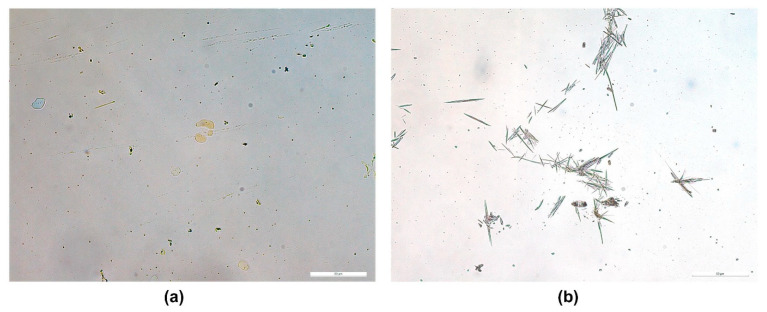
Brightfield microscopic image of (**a**) separation of drug at 5% *w*/*w* from dried silicone adhesive and (**b**) crystal formation in dried acrylate adhesive. The scale bar represents 50 µm. Reprint with permission from Ganti et al. [90], Elsevier, 2023.

**Figure 8 pharmaceutics-15-02577-f008:**
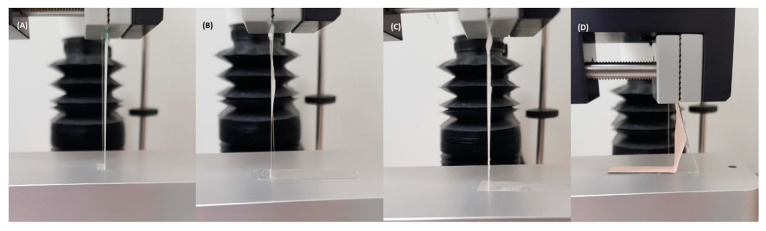
Peeling behaviours of PSA-backing membrane combinations during the 90° peel adhesion test. (**A**) Cotran™ 9720–TR4, (**B**) Scotchpak™ 9733–TR4, (**C**) Scotchpak™ 9735–TR4 and (**D**) Scotchpak™ 9730–DB3. Reprint with permission from Kim et al. [51], Elsevier, 2023.

**Table 2 pharmaceutics-15-02577-t002:** Microneedle patches for the treatment of non-melanoma skin cancer.

Therapeutic Agent	Materials Used for Microneedle Preparation	Incorporated Nanosystem	Skin Cancer Model	Key Findings	Ref.
5-Aminolevulinic acid (5-ALA)	Stainless steel	N/A	A20 tumour-bearing Balb/cA nude mice	Stainless steel microneedles coated with 5-ALA achieved much deeper skin penetration (~480 µm) compared to its topical cream counterpart (~150 µm). The microneedles significantly reduced subcutaneous tumour growth by about 57%. Conversely, the topical cream containing 5-ALA (5 mg) failed to suppress the tumour volume.	[60]
Gold	Poly(L-lactide)	PEGylated Gold Nanorod	Female A431 tumour-bearing Balb/cA nude mice	The GNR-PEG@MN demonstrated excellent skin penetration capabilities with a height of 480 μm whilst achieving effective heat transfer in vivo, with the tumour sites reaching 50 °C within 5 min. The combination of low-dose MPEG-PDLLA-DTX micelles and GNR-PEG@MNs eliminated the A431 tumour in vivo with no recurrence.	[61]
5-FU Indocyanine green (ICG)	Hyaluronic acid	Monomethoxy-poly (ethylene glycol)-polycaprolactone nanoparticle	A431 tumour-bearing Balb/cA nude mice	5-Fu-ICGMPEG-PCL@HA MN demonstrated a good skin penetration of 600 µm with a rapid heating transfer efficacy to 60 °C in 5 min upon 808 nm near-infrared laser. The microneedles also demonstrated tumour inhibition capability without recurrence.	[17]
Cisplatin	Biocompatible photopolymer resin	N/A	A431 human squamous carcinoma xenografts in Balb/c nude mice	The microneedles were fabricated using biocompatible photopolymer resin via stereolithography 3D printing. Cisplatin was coated on the needle surface via inkjet printing. 3D-printed microneedles had good skin penetration, achieving 80% penetration depth (737.7 ± 63.7 µm). Rapid cisplatin release rates (80–90%) were observed in the first 1 h. In vivo evaluation demonstrated that the cisplatin permeated sufficiently, exhibiting high anticancer activity and resulting in 100% tumour regression.	[62]
IMQ	Polyvinylpyrrolidone and vinyl acetate (PVPVA); Polyethylene glycol 400	N/A		Microneedles loaded with IMQ, utilising a polyvinylpyrrolidone-co-vinyl acetate polymer achieved a penetration depth of 426 ± 72 µm. Despite the microneedle containing an IMQ load six times lower than the clinical dose of Aldara^®^, it achieved a similar level of IMQ intradermal delivery.	[63]

N/A = Not applicable.

**Table 3 pharmaceutics-15-02577-t003:** Drug-in-adhesive and matrix-type patches for potential non-melanoma skin cancer treatment.

Patch Type	Therapeutic Agent	Materials Used for Patch Preparation	Incorporated Nanosystem	Key Findings	Ref.
Matrix (referred to as polymeric by the authors)	IMQ	PMVE/MA, tripropyleneglycol methyl ether	N/A	Polymeric patches containing IMQ of 4.75, 9.50, and 12.50 mg cm^−2^ were developed. The patches released significantly more drug through a Cuprophan^®^ dialysis membrane than the commercial cream, Aldara^®^ over 6 h.	[18]
Matrix (referred to as polymeric by the authors)	Gold	Gantrez^®^ S-97 (copolymer of methyl vinyl ether and maleic acid, Mw = 1,200,000	Functionalised gold nanorods with thiolated poly(ethylene) glycol	Polymeric film containing gold nanorods (GnRs) for use in local hyperthermia applications was developed. The GnR-loaded films were able to heat the skin model over 40 °C, demonstrating its potential for non-invasive local hyperthermia applications against NMSC.	[68]
Drug-in-adhesive (DIA)	5-FU	Dimethylaminoethyl methacrylate, butyl methacrylate and methyl methacrylate (2:1:1) (Eudragit^®^ E)	N/A	DIA patches containing 5-FU were developed for the first time using Eudragit^®^ E as an adhesive polymer matrix. The patches containing 40% (relative to the polymer ratio) triethyl citrate, dibutyl sebacate, or triacetin as a plasticiser achieved adhesive properties similar to several marketed patches. A controlled release of 5-FU was achieved, suggesting its potential application in skin cancer treatment.	[51]
DIA	5-FU and IMQ	Eudragit^®^ E	N/A	DIA patches using Eudragit^®^ E and triacetin were developed containing both 5-FU and IMQ. The in vitro release rate of 5-FU was quicker than that of IMQ, with about 75% of the drug content released within the initial 50 min and 120 min, respectively.	[69]
Matrix	IMQ	hydroxypropyl methylcellulose (HPMC) K4M, propylene glycol	Nanostructured lipid carriers (NLCs)	NLCs containing IMQ were developed by Design of Experiments. The optimised formulation was then incorporated into a matrix-type topical patch consisting of hydroxypropyl methylcellulose (HPMC) K4M and propylene glycol. The ex vivo deposition study demonstrated that the IMQ-NLCs patch significantly increased IMQ deposition in the deeper skin layers than Aldara^®^ cream.	[20]

## Data Availability

Not applicable.

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
