# Peer review of "Innovative Topical Patches for Non-Melanoma Skin Cancer: Current Challenges and Key Formulation Considerations"

_pharmaceutics, 2023, doi:10.3390/pharmaceutics15112577_

Round 1

Reviewer 1 Report

Comments and Suggestions for Authors

Congratulations for the work done. It is very well structured and easy to understand. Although I consider that a brief section on hydrogel patches should be included, which include the nanomaterials most used for the treatment of this type of pathology. Since they are referred to in lines 213-215 and are subsequently not included in any section of the work.

I would like to make a few formatting suggestions:

1. Figure caption 5 should be specified or briefly explained.

2. Section 7 appears as 7.7, it is an error.

3. Adapt the format of bibliographic references to the instructions of the journal: name of the journal abbreviated with period and italics, year in bold and volume in italics.

Thank you very much in advance and again, congratulations on the work.

Reviewer 2 Report

Comments and Suggestions for Authors

It was a review study discussing the application of innovative topical patches used for the treatment of non-melanoma skin cancer. There are some tiny comments on this study that should be considered before publication:

1-      The last paragraph of the introduction needs to be rewritten.

2-      What do “BCC and SCC” refer to? All the abbreviations should be introduced at their first usage.

3-      Please check all the figures, some of them were not referred to in the main text.

Comments on the Quality of English Language

There are some grammatical mistakes in the text that should be corrected. Some of them are as follows:

-          The first stainless steel microneedles with 5-Aminolevulinic acid (5-ALA) as a photosensitiser using a micro-precision dip coater were presented

-          In vivo animal studies conducted on female Balb/C mice with A20 cancer cells, microneedles coated with 5-ALA significantly suppressed the subcutaneous tumour growth by about 57%.

-          … microneedle patch using coated with a near-infrared light-responsive monomethoxy-poly (ethylene glycol)-polycaprolactone NPs …

Reviewer 3 Report

Comments and Suggestions for Authors

Dear Authors,

I have reviewed your paper with interest, and I believe it is attractive and present novelties.

Line 178: I agree with you, but as you know destruction of the premalignant cells is necessary for cured them, and thus, some side-effects will appear anyway. Could you discuss deeply this point.

Line 234: About microneedle array patches, maybe you can add a discussion about an important issue, that is that not all topical treatment can be delivered by this path or way, and why.

Figure 4: It is very interesting and well represented.

In my view the conclusions are very long, and the text would be more legible if it was shortened.

Reviewer 4 Report

Comments and Suggestions for Authors

This review manuscript begins with an overview of Non-melanoma skin cancer (NMSC) as well as the current landscape of topical treatments for NMSC, specifically focusing on the emerging technology of topical patches. This work unprecedentedly combines and discusses all the current advancements in innovative topical patches for the treatment of NMSC. The authors present the key considerations and emerging trends in the construction of these advanced topical patches. The manuscript is quite interesting and the scientific rigor is adequate for the journal. The manuscript is well-structured and my suggestion is to publish this work in this journal after minor revisions according to specific comments.

 1.      A review manuscript not only exposes the active ingredients present on the commercial market. In this sense, I suggest the authors reflect in the manuscript the recent advances in active ingredients (non patches techonology) for topical therapy of NMSC such as 5-Aminolevulinic acid, doxorubicin, metformin, carvedilol, paclitaxel, retinoids as well as active ingredients of natural origin such as curcumin, and photodynamic therapy…to help have a global view of the content

2. I would suggest to the authors in this review manuscript a specific section for NMSC epidemiology.

3.      It is evident that the manuscript presents recent advances in innovative topical patches for the treatment of NMSC. I also suggest to the authors reflect in the introduction parts before the detailed description of patches technology, most of numerous technologies, such as nanoparticles, including polymeric nanoparticles, liposomes, gels, nanogels, transferosomes, micelles, dendrimers, niosomes for topical delivery of anticancer products apart from the topical patch technology.

4.      In the manuscript the authors classify the promising innovative topical patches for treatment of NMSC: microneedles, polymeric patches and hydrogels. The first two types are widely exposed but there is no section of hydrogels. Why? I suggest to the authors a section with the main advances in Topical Patch Technology for hydrogels for NMSC treatment just as they have done with microneedles and polymeric patches.
